# Linking Adult Olfactory Neurogenesis to Social Reproductive Stimuli: Mechanisms and Functions

**DOI:** 10.3390/ijms26010163

**Published:** 2024-12-28

**Authors:** Serena Bovetti, Sara Bonzano, Federico Luzzati, Claudio Dati, Silvia De Marchis, Paolo Peretto

**Affiliations:** 1Department of Life Sciences and Systems Biology, University of Turin, Via Accademia Albertina 13, 10123 Turin, Italy; serena.bovetti@unito.it (S.B.); sara.bonzano@unito.it (S.B.); federico.luzzati@unito.it (F.L.); claudio.dati@unito.it (C.D.); 2NICO—Neuroscience Institute Cavalieri Ottolenghi, University of Turin, Regione Gonzole 10, 10043 Orbassano, Italy

**Keywords:** accessory olfactory bulb, main olfactory bulb, vomeronasal system, pheromones, brain plasticity, multisensory integration

## Abstract

Over the last three decades, adult neurogenesis in mammals has been a central focus of neurobiological research, providing insights into brain plasticity and function. However, interest in this field has recently waned due to challenges in translating findings into regenerative applications and the ongoing debate about the persistence of this phenomenon in the adult human brain. Despite these hurdles, significant progress has been made in understanding how adult neurogenesis plays a critical role in the adaptation of brain circuits to environmental stimuli regulating key brain functions. This review focuses on the role of olfactory neurogenesis in the brain’s response to social reproductive cues in rodents, highlighting its influence on animal behaviors critical for survival. We also address open questions and propose future directions to advance our understanding of the relationship between adult neurogenesis and reproductive function regulation.

## 1. Introduction

The discovery that the adult mammalian brain can generate and integrate new neurons—a process known as adult neurogenesis—has been a groundbreaking development in neuroscience since the early 1990s. In 1993, studies by M. Luskin and Alvarez-Buylla [1,2] identified neuronal progenitors in the subventricular zone (SVZ) of the forebrain of adult rats and mice that give rise to neuroblasts that migrate to the olfactory bulb (OB), where they generate large numbers of inhibitory interneurons in the granule (GCL) and glomerular (GL) cell layers. These findings revived earlier data by J. Altman (1962), M. Kaplan (1977), and S. Bayer (1983) [3,4,5] on the existence of postnatal neurogenesis. Subsequent findings identified neural stem cells (NSCs) in both the SVZ and the subgranular zone (SGZ) of the dentate gyrus (DG) in the hippocampus, establishing two distinct “neurogenic niches” within the adult mammalian brain [6]. Additionally, compelling evidence indicates that the adult mammalian hypothalamus is also a neurogenic area, capable of producing new neurons in physiological conditions, albeit at a much lower rate than the SVZ-OB and the DG [7].

These discoveries challenged the long-standing dogma that neurogenesis occurs only during embryonic and early postnatal stages and raised expectations about the potential of endogenous NSCs for therapeutic applications in neurodegenerative diseases. Despite extensive research, efforts to harness adult NSCs for regenerative purposes have yielded limited results [8], and the existence and extent of adult neurogenesis in humans remains a topic of ongoing debate [9]. Nevertheless, findings in animal models, particularly in rodents, have shed light on the physiological role of adult neurogenesis, and have advanced our understanding of how mature brain circuits adapt to environmental stimuli through the integration of new neurons [10]. For example, hypothalamic neurogenesis is thought to function as an adaptive mechanism in response to metabolic changes driven by environmental or internal conditions [11,12]. In the DG of the hippocampus, adult-born excitatory granules contribute to learning and memory as well as affective behaviors [13]. On the other hand, granule and periglomerular olfactory adult-born interneurons play a role in the processing of chemosensory information and the formation of odor memory [14]. Moreover, several studies have demonstrated the implication of adult neurogenesis in the processing of social stimuli relevant to reproductive behaviors (see, for example, [15,16]). For instance, in female mice, reproductive behaviors such as mate choice and pup care elicited by social stimuli, rely on the integration of pools of new neurons in both the main OB (MOB) and accessory OB (AOB) [17,18,19,20,21,22]. Adult neurogenesis in these areas is finely modulated by a combination of external (i.e., pheromones) and internal (i.e., reproductive hormones) stimuli. Together, these highly integrated processes regulate the activity of downstream nuclei involved in the control of reproductive functions [19,23,24]. Unraveling the mechanisms underlying these processes provides unique insights into how neural plasticity adapts the brain in response to salient environmental signals. In this review, we discuss findings and outstanding questions concerning the relationship between adult olfactory neurogenesis and the regulation of reproductive functions, highlighting areas for future research.

## 2. Anatomical Insights into the Role of Adult Neurogenesis in Reproductive Behavior

The first evidence linking adult neurogenesis to social reproductive stimuli emerged with the discovery that neuroblasts originating in the SVZ and migrating towards the OB also reach the AOB in adult rats [25]. The AOB is a spheroidal nucleus integrated in the dorsal region of the main OB that receives inputs from the vomeronasal organ (VNO), a chemosensory epithelium specialized to detect pheromones, chemical signals associated with social communication (Figure 1A,B) [26]. Functionally, the AOB acts as the first relay station of the vomeronasal system (VNS), a subcortical pathway that includes key nuclei, such as the medial amygdala (MeA), involved in modulating reproductive behavior, through multisensory integration of social cues [27]. Subsequent studies, using lineage tracing of homotopically transplanted EGFP + SVZ progenitors, combined with immunofluorescence and 3D reconstruction analysis, demonstrated that as for the MOB, neuroblasts migrating to the AOB in both male and female mice acquire distinct positioning and neurochemical properties typical of functionally integrated granule and periglomerular interneurons [28]. Given the unique functional properties that newborn neurons bring to pre-existing circuits [29,30], along with the privileged position of the AOB as a gate for pheromonal signals within the VNS, intriguing questions have been raised about the potential role of adult neurogenesis in regulating social and reproductive behavior.

## 3. Activity-Dependent Regulation of Adult Neurogenesis and Its Function in the Context of Social Reproductive Stimuli

The survival of newborn neurons in the MOB circuits has been shown to be regulated through activity-dependent processes. For instance, data obtained through sensory deprivation (e.g., olfactory mucosa lesion or naris occlusion) have shown a negative modulation on the number of functional newborn neurons integrated into the OB circuits [31,32,33]. On the other hand, studies exposing animals to olfactory enrichment (i.e., long-term exposure to a rich range of odors) have demonstrated that environmental cues can significantly increase the survival rate of newborn neurons integrated into the main OB layers [34,35] and transiently improve odor memory in mice [36], suggesting a role for adult neurogenesis in olfactory memory.

A key indicator that adult neurogenesis could be functionally linked to reproduction has come from results showing its modulation by the same environmental stimuli that elicit reproductive behaviors. Exposure of female mice to adult male bedding (i.e., containing semiochemicals present in urine and exocrine gland secretion) [37] was found to significantly increase the survival of newly generated neurons integrating into AOB circuits [28]. Remarkably, this pro-neurogenic effect was only found in sexually mature females and not in prepubertal females or males [18,19,23,24,28,38], suggesting that it may depend on several underlying biological factors related to hormonal differences, reproductive biology, and sex-specific brain plasticity. Furthermore, using c-fos expression as a marker of neuronal activity, it was shown that shortly after their integration into female AOB circuits, newborn neurons are transiently more responsive to familiar (i.e., previously experienced) male pheromones than to unfamiliar ones. This transient role is likely due to the higher excitability, lower threshold for synaptic and structural plasticity, and increased responsiveness to experienced stimuli that are typical of young newborn neurons [29,30,39]. This observation supported the hypothesis that the continuous addition of newborn neurons to the AOB may provide a preferential substrate for the formation of transient memories of previously experienced male individual signatures [18].

Definitive evidence demonstrating the functional role of adult neurogenesis in reproduction has come from experiments involving the depletion of newborn cells through different methods (e.g., by Ara-c injection, X-ray irradiation—gene-coded selective deletion). These approaches have shown that certain reproductive behaviors, such as mate recognition [18] or parental care [20,21], require the integration of a pool of newborn neurons in the OB. In mate recognition, for example, the integration of a pool of newborn neurons in the AOB triggers the formation of a transient memory for the mating partner (mate pheromonal imprinting), that is necessary to avoid the so-called Bruce effect [18], which consists in a neuroendocrine reflex leading to a pregnancy block following female exposure to unfamiliar male pheromones, during a critical postmating window [40]. Notably, a pro-neurogenic effect of male pheromones in female mice was also identified in the MOB, consistent with the complementary roles of both the main and vomeronasal olfactory systems in the perception of social stimuli [41]. Accordingly, multiple studies in different mammalian species have shown a positive modulation of adult neurogenesis in both the AOB and MOB during intersexual and parent-offspring interaction [17,21,22,42,43,44,45].

Social reproductive cues not only promote the survival of newborn neurons in the OB but also enhance the proliferation of neural progenitors in the SVZ and in the DG of hippocampus [17,43,46]. A recent study by Chaker and colleagues [22] confirmed that pregnancy promotes neurogenesis in the OB region through increased proliferation of SVZ progenitors and added intriguing evidence indicating that this physiological state activates spatially distinct pools of NSCs in the maternal SVZ, resulting in the integration of specific newborn interneuron subtypes into the AOB and the MOB sub-layers, where fewer newborn neurons are typically added under non-pregnant conditions. The integration of these neurons is proposed to be necessary for parental care activities, including the recognition of one’s own pups. As in mate recognition [18], newborn interneurons generated during pregnancy are functionally recruited (i.e., exhibiting preferential c-fos expression) in a specific time-window that, in the case of parental care, corresponds to the pre-weaning period. This supports the idea that adult neurogenesis in the OB provides pools of “young and excitable” neurons on demand to cope with specific temporally defined functions, modulating different aspects of reproductive behaviors.

## 4. The Interplay Between External and Internal Reproductive Cues Regulating Adult OB Neurogenesis

In female mice, exposure to male pheromones not only enhances adult neurogenesis but also stimulates the hypothalamic–pituitary–gonadal (HPG) axis, which releases secretory factors and gonadal hormones that control reproduction [47]. These sex hormones, in turn, modulate adult neurogenesis based on age, sex, individual experience, and the overall internal state of the body [23,48,49].

Thus, a fine-tuned modulation of adult neurogenesis through the integration of external (pheromones) and internal (sexual hormones) cues seems relevant to optimize reproductive behaviors (Figure 1C). To gain insight into these complex regulatory mechanisms, various animal models have been exploited [24]. Studies in Semaphorin 7A knockout (Sema7A KO) mice—which exhibit reduced numbers of gonadotropin-releasing hormone (GnRH) neurons, small testes, and subfertility [50]—and in wild-type males castrated in adulthood have shown that the level of circulating testosterone in male mice is crucial for the sex-specific regulation of AOB neurogenesis [19]. In males characterized by low testosterone circulating levels (e.g., Sema7A KO and/or adult gonadectomized males), exposure to male pheromones increases the integration/survival of newborn interneurons in the AOB, a phenomenon that is normally restricted to females [18]. This, in turn, affects the pattern of neuronal activity in the downstream nuclei of the VNS, which influences opposite-sex cue preference and attraction [19]. Strikingly, in adult Sema7A KO males, chronic testosterone treatment was shown to be sufficient to reverse the feminized responses to male pheromone exposure in terms of AOB neurogenesis, c-fos expression pattern along the VNS pathway, and appetitive behavior [19]. These data clearly indicate that the sexually dimorphic integration of newborn neurons elicited by male pheromones in the AOB of adult mice involves a gonadal-dependent mechanism. They also support that the gonadal-dependent integration of newborn neurons can influence the activity of the VNS nuclei involved in the control of sexual behavior.

Another striking example of the interplay between HPG-axis hormones and the sex-specific modulation of adult neurogenesis comes from research on a transgenic GnRH-deficient mouse model (GnRH::Cre;Dicer^loxP/loxP, here referred to as GnRH::CreDicer KO; [23]. Beginning at puberty, GnRH production and secretion orchestrate HPG-axis activity and are essential for reproductive function [51]. In the GnRH::CreDicer KO mice, the enzyme Dicer—an RNAse-III endonuclease essential for miRNA biogenesis [52]—is selectively inactivated in GnRH neurons. As a result, these mice fail to undergo puberty and exhibit severe hypogonadism and sterility in adulthood [50]. Importantly, in these mice, GnRH production gradually declines in the infantile period, affecting only the onset of puberty [50], without disrupting the perinatal, endocrine-dependent organizational phase of brain development [53,54].

According to data indicating puberty as a critical stage of life characterized by increased secretion of gonadal hormones [55] and refinement of neural circuits that drive reproduction [56], the alteration of the hormonal milieu found in the GnRH::CreDicer KO line leads to sex-specific dysregulation of adult OB neurogenesis [23,24]. In males, there is a reduction in proliferative progenitors (identified by Ki67 expression), including neuronal-lineage-committed cells (Ki67+/DCX+ cells), restricted to the dorso-lateral subdomain of the SVZ. This suggests that, in males, only a subset of olfactory newborn interneuron progenitors is affected by the loss of peripubertal HPG-axis hormones. In GnRH::CreDicer KO females, a decrease in the survival of newly generated cells was observed in the MOB but not in the AOB. This effect was also found in females who had undergone an ovariectomy shortly before puberty (at postnatal day 21), supporting the idea that, among the different HPG-axis factors, the alteration in cell survival found in females can be mostly attributable to the loss of gonadal steroids. Interestingly, in the DG of both male and female GnRH::CreDicer KO mice, no alteration in neuronal progenitor proliferation, neuronal specification, or newborn cell survival was observed. These findings indicate that HPG-axis secretory activity at puberty selectively impacts the SVZ-OB neurogenic system, emphasizing the importance of puberty in setting a sexually dimorphic regulation of the OB neurogenic process in adulthood.

## 5. Advancing Our Understanding of Adult Neurogenesis in Reproductive Behavior

### 5.1. Multisensory Integration of Reproductive Cues and Adult Neurogenesis

In addition to pheromones, other environmental cues conveyed through the auditory and tactile systems contribute to the complex regulatory mechanisms underlying reproductive responses and may influence adult olfactory neurogenesis, either directly or through synergistic effects with olfactory cues (Figure 1C). In mate choice and pup recognition behavior—both mediated by OB neurogenesis through pheromone perception [17,18,22]—a possible synergistic contribution of acoustic stimuli to neurogenesis regulation is conceivable. Indeed, in addition to pheromones, ultrasonic vocalizations (USVs) emitted by males are among the most important cues used by female mice in mate choice behavior [57]. Interestingly, USV-induced attraction to a specific male is enhanced in the presence of male pheromones, such as the exocrine gland-secreting peptide 1 (ESP1), a key signal responsible for the Bruce effect [58], which is modulated by adult neurogenesis [18]. Similarly, maternal pup retrieval is regulated by multisensory information from pup olfactory and auditory cues, and exposure to pup odors enhances neuronal responses to pup USVs in the maternal auditory cortex [59]. These findings support the salience of acoustic stimuli to post-pregnant dams and suggest their possible involvement in the regulation of the transient pools of neonatal OB neurons, which have recently been described to be involved in promoting parental care [22]. Previous studies have already indicated a potential role for auditory cues in regulating adult neurogenesis. Exposure to diverse and novel acoustic landscapes, including non-ethologically relevant auditory cues for mice such as Mozart and silence, has been shown to promote cell proliferation in the adult female mouse DG [60]. Additionally, lower levels of cell proliferation in the adult rat hippocampus have been correlated with inner ear cell loss [61]. These findings suggest that auditory cues are among the environmental stimuli capable of modulating adult neurogenesis. Future research should explore whether this modulation also occurs for SVZ neurogenesis and how multimodal stimuli, such as olfactory and acoustic cues, may integrate to regulate OB adult neurogenesis in the reproductive context.

Physical contact (i.e., touch) may be also involved in modulating adult neurogenesis in the OB within the reproductive context. There is ample experimental evidence supporting this hypothesis. Direct physical contact promotes the exploration of olfactory reproductive cues, activating both the main and vomeronasal olfactory systems [41]. The activation of the vomeronasal pathway, which involves direct contact with the non-volatile components of pheromones, is essential for the pheromone-dependent modulation of neurogenesis in the AOB [18,28]. During mating, vaginal stimulation in female mice induces the release of noradrenaline from the locus coeruleus in the AOB, promoting the formation of a mate memory [62], which involves the integration of new neurons in the AOB [18]. Other studies have demonstrated the importance of top-down noradrenergic signals to promote olfactory perceptual learning, which requires the incorporation and activation of newborn neurons in the OB [63,64]. Modulation of adult neurogenesis through physical social interaction also occurs during pace mating behavior [44,65]. In natural conditions, rodent females can control the pacing of sexual interaction, thereby managing the frequency and intensity of the sexual stimulation they receive. This induces a reward state that increases neurogenesis in the OB, possibly via top-down signals [66].

Which neural circuits regulate the possible multimodal impact on adult olfactory neurogenesis? A potential pathway involves top-down control from hub regions known to integrate multisensory information. Among these regions, the amygdala (particularly the medial and basolateral amygdala) may be a key candidate for controlling olfactory neurogenesis, as it processes sensory information from both the olfactory and vomeronasal systems [67,68]; it is reciprocally connected with the AOB [69], and links to cortical regions such as the medial prefrontal cortex, which is activated by pheromones and auditory cues (Figure 1A) [70]. Moreover, it is also involved in the neural control of affiliative touch in prosocial interaction in social species, including humans [71], through reciprocal connections with several cortical regions involved in processing of touch stimuli [72]. Interestingly, an excitotoxic lesion in the medial amygdala blocks the pheromone-dependent increase integration of newborn neurons in the AOB [18]. Finally, the MeA is part of the vomeronasal pathway, which impacts the HPG-axis release of neuroendocrine factors (internal cues) known to modulate adult neurogenesis. Overall, these data highlight the central role of the amygdala in the complex interplay between external and internal cues eliciting reproduction through modulation of OB neurogenesis (Figure 1).

It is noteworthy that, as mentioned above, exposure to social stimuli released during sexual and parent–offspring interactions, in addition to the SVZ-OB, triggers neurogenesis in the DG niche [17], possibly also through a reciprocal connection between the olfactory system and the hippocampus [73]. As social interactions are multisensory, modulation of hippocampal neurogenesis may be mediated by other sensory channels in addition to olfaction. For example, in rats, a specific tactile stimulation, tickling, promotes proliferation in the hippocampus when the animal emits a high number of 50 kHz calls, i.e., a condition related to an appetitive state mediated by affective behavior [74]. During pregnancy and the postpartum period, there are large fluctuations in the concentrations of reproductive hormones (e.g., estrogen, oxytocin, progesterone, and prolactin) that correlate with cognitive and behavioral changes associated with parental care. Some of these hormones not only influence adult neurogenesis in the SVZ-OB but also act as potent modulators of neurogenesis in the hippocampal niche [75] and in the hypothalamus [49]. Enhanced neurogenesis at these sites may contribute to learning, memory, mood, and other neurophysiological aspects involved in reproductive activities.

### 5.2. Identification of Neuronal Progenitors Activated by Social-Reproductive Cues

Another aspect of OB neurogenesis that requires further clarification is the identification of SVZ progenitors that contribute to the pools of newborn neurons involved in the modulation of social–sexual stimuli. It is not known whether there are separated SVZ microdomains and/or progenitors that contribute selectively to the genesis of the interneurons destined to the MOB versus the AOB. Moreover, it is unclear whether a specific subset of SVZ progenitors selectively generates neurons recruited into OB circuits in the context of social reproductive functions.

Adult SVZ progenitors become regionally specified early in development [76], keeping a mosaic organization into SVZ specialized domains that generate distinct OB interneuron subtypes [76,77]. The activity of these progenitors and their exit from quiescence result from a fine-tuned integration of internal and external cues that influence the complex microenvironment of the neurogenic niche [78]. The analysis of SVZ neural stem cell proliferation during pregnancy has revealed multiple spatially and temporally restricted domains, which increases cell division, particularly in regions that are normally either poorly active, such as the ventral domain, or completely inactive, such as the medio-dorsal domain [22]. Further studies are needed to clarify whether the progeny of these activated progenitor domains corresponds to the pool of newborn neurons engaged during parental care behavior, as well as to identify the molecular mediators and cellular mechanisms involved in activating these specific domains. Among pregnancy hormones, prolactin has been previously shown to activate SVZ progenitor proliferation, representing a likely molecular candidate [46,79], but it is not yet known whether and how prolactin specifically impacts on the ventral and medio-dorsal SVZ domains. Moreover, the SVZ receives projections from different brain circuits, including the dopaminergic, serotonergic and cholinergic systems known to be implicated in social behaviors [78], raising the intriguing possibility of spatially confined effects regulated by specific projections. Understanding how different progenitor domains integrate local and long-range stimuli to regulate adult neurogenesis in various social and reproductive functions could be key to uncovering its connection to behavior.

### 5.3. Future Perspectives on Adult Neurogenesis in the Human Brain

As mentioned earlier, interest in the field of adult neurogenesis has declined in recent years in relation to the difficulty of extrapolating translational data. Nevertheless, comparative studies have shown that although there are species–specific differences, the process of adult neurogenesis is an evolutionarily conserved feature in most mammals [80]. Results referring specifically to primates and, among them, to humans indicate that, in the olfactory region, the migration of newborn neurons from the SVZ ceases early in the juvenile phase [81]. On the other hand, some studies have provided evidence in support of the occurrence of adult neurogenesis in the human hippocampus and striatum [82], but this phenomenon remains highly debated, not only in terms of its presence and/or extent but also in functional terms, i.e., to what extent should neurogenesis contribute (how many neurons should integrate) to have an impact at the functional level [83]. Despite the technical and ethical limitations associated with studying such processes in the human brain [9], given the comparative data on the complex and integrated regulatory mechanisms underlying adult neurogenesis, some points still need to be considered in support of further investigations in this field of research. As demonstrated in animal models, the activity and modulation of adult neurogenesis is strongly influenced by endocrine factors [24]. It is therefore possible that a specific hormonal milieu, such as that characterizing pregnancy and the parental care phase, might stimulate/reactivate quiescent progenitors also in the adult human SVZ [22]. As a matter of fact, olfactory function is relevant to reproductive behavior in humans, though a clear picture of the mechanisms underlying such function is still lacking [84].

The human SVZ is considered inactive in adult life but, within it, are contained competent progenitors capable of activation in vivo following stroke [85]. Similarly, in the mouse striatum, as in other mammalian species, there are quiescent glial progenitors that produce transient populations of newborn neurons following injury [86]. Thus, it is possible to envisage that, under specific hormonal conditions, some SVZ progenitors might activate and originate new neurons contributing to transient functions even in the human brain. In addition, recent data obtained in various regions of the telencephalon of several mammals, including primates and among them Homo sapiens, support the existence of ‘immature neurons’ [80] that could potentially represent a reservoir of cells ready to differentiate and integrate on demand in response to specific stimuli. Given the transient roles played by immature neurons during adult neurogenesis, particularly in social and reproductive behaviors, it would be interesting to investigate the possible involvements of such “immature neurons” in these contexts.

## 6. Conclusions

After more than 30 years of research on adult neurogenesis, the abundant evidence for the integration of new neurons into existing circuits to support specific brain functions, in particular the mediation of reproductive behavior, strongly supports its relevance in mammals. However, as discussed in the previous sections, there are many unanswered questions about adult neurogenesis that deserve further investigation in both animal models and humans. It is therefore envisaged that the use of a combination of different advanced and cutting-edge techniques will provide the field with new answers to refine our understanding of how adult neurogenesis contributes to brain function. Specifically, with respect to reproductive behavior, the use of spatial transcriptomic gene expression at single cell resolution may help to unravel the identity of neural progenitors and pools of newborn neurons recruited by multisensory exposure to salient cues associated with different reproductive behaviors. In parallel, the use of techniques to image neuronal circuits activation in toto (e.g., whole-brain light sheet microscopy coupled with c-fos immunofluorescence), following multisensory stimuli, could improve our knowledge of the circuits activated by salient cues and the specific contribution of newborn neurons through comparisons with models of neurogenesis depletion. As our knowledge of adult neurogenesis in animal models grows, new insights into possible fine-tuning mechanisms of adult neurogenesis will help to refine the scientific questions and focus research on adult human neurogenesis.

## Figures and Tables

**Figure 1 ijms-26-00163-f001:**
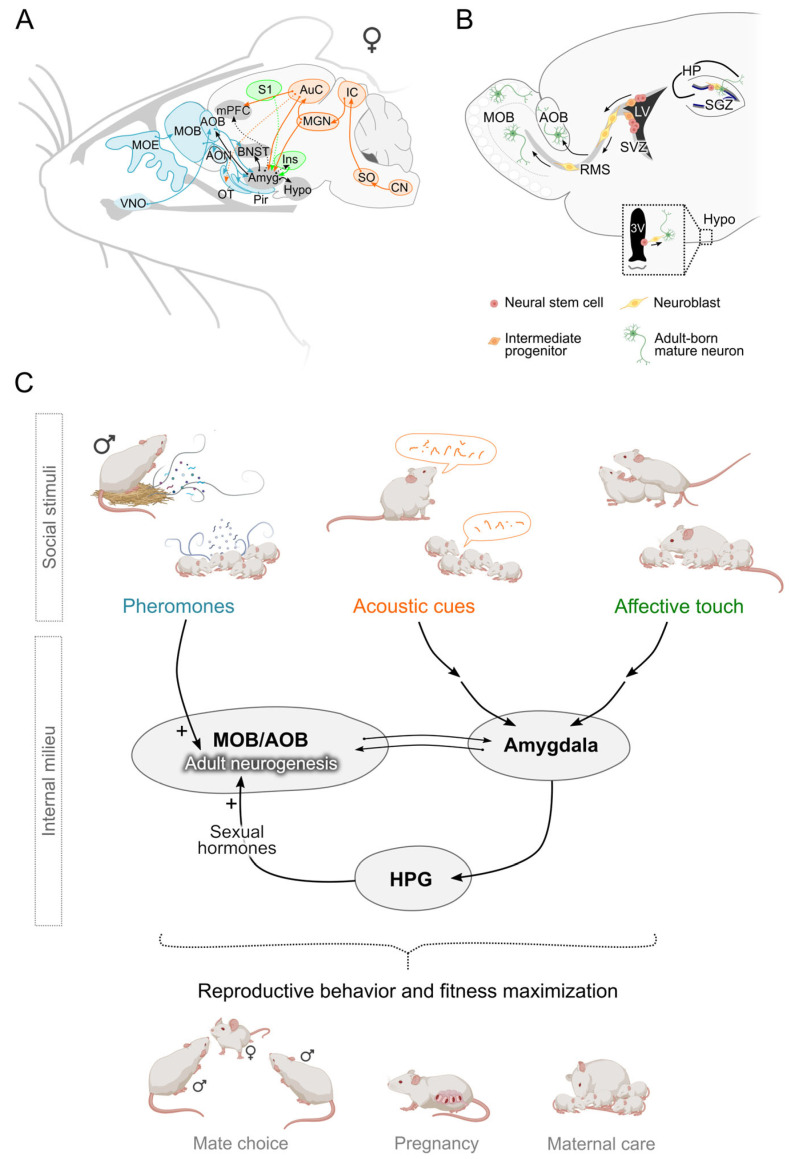
(**A**). Schematic of female neural pathways for social stimuli including chemical, acoustic, and tactile signals. The chemosensory pathway is shown in light blue, the auditory pathway in orange, the main areas involved in touch in green, the central hub areas in gray; arrows represent connections between regions, dotted lines represent pathways possibly involved in multisensory integration. (**B**). Schematic of constitutive adult neurogenesis in the mouse brain. Arrows indicate directions of neuroblast migration from site of origin to site of integration (**C**). Proposed model for the multimodal integration of social stimuli crucial for shaping reproductive behavior and maximization of fitness through modulation of adult OB neurogenesis. Arrows indicate functional connections among brain regions involved in the processing of social stimuli. Abbreviations: accessory olfactory bulb (AOB), amygdala (Amyg), anterior olfactory nucleus (AON), auditory cortex (AuC), bed nucleus of the stria terminalis (BNST), cochlear nucleus (CN), hippocampus (HP); hypothalamic–pituitary–gonadal axis (HPG), hypothalamus (Hypo), inferior colliculus (IC), insula (Ins), lateral ventricle (LV), medial geniculate nucleus (MGN), main olfactory bulb (MOB), main olfactory epithelium (MOE), medial prefrontal cortex (mPFC), olfactory tubercle (OT), piriform cortex (Pir), rostral migratory stream (RMS), somatosensory cortex (S1), subgranular zone (SGZ), subventricular zone (SVZ), superior olivary nucleus (SO), vomeronasal organ (VNO), third ventricle (3V). Cartoons in the figures have been created with Biorender.com.

## Data Availability

No new data were created or analyzed in this study. Data sharing is not applicable to this article.

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
