# Peer review of "Linking Adult Olfactory Neurogenesis to Social Reproductive Stimuli: Mechanisms and Functions"

_ijms, 2024, doi:10.3390/ijms26010163_

Round 1
Reviewer 1 Report
Comments and Suggestions for Authors
The manuscript by Bovetti et al provides a useful review of the recent advances on the possible roles that adult neurogenesis may play on social reproductive functions. It is scholarly written, I found it very easy to read and it provides, to my knowledge, most of the relevant references in the field.
I have no major comments, and I found only one typo in Fig 1C ("Amydgala" should be "Amygdala").
I should congratulate the authors for their work.
Author Response
Comments: The manuscript by Bovetti et al provides a useful review of the recent advances on the possible roles that adult neurogenesis may play on social reproductive functions. It is scholarly written, I found it very easy to read and it provides, to my knowledge, most of the relevant references in the field.I have no major comments, and I found only one typo in Fig 1C ("Amydgala" should be "Amygdala"). I should congratulate the authors for their work.
Response: We would like to thank the reviewer for his positive comments on the review and for pointing out the spelling error in the figure, which has now been corrected in the revised version of the manuscript.
Reviewer 2 Report
Comments and Suggestions for Authors
The review “Linking Adult Neurogenesis to Social Reproductive Stimuli: Mechanisms and Functions” is interesting and covers different topics, especially those in which the authors have several publications. I only have a couple of suggestions that the authors need to include, especially the one describing a neurogenic site in the hypothalamus.
1.- In the introduction, the authors described the 2 classic neurogenic niches: The svz olfactory bulbs and the SVZ/SGZ of the dentate gyrus of the hippocampus. However, evidence also suggests that the hypothalamus is another site of neurogénesis. This third site needs to be included in the review. Here are some references about the topic.
Hypothalamic Subependymal Niche: A Novel Site of the Adult Neurogenesis. Ewa Rojczyk-Gołe˛biewska, Artur Pałasz, Ryszard Wiaderkiewicz
Cell Mol Neurobiol (2014) 34:631–642 DOI 10.1007/s10571-014-0058-5
Hypothalamic Neurogenesis as an Adaptive Metabolic Mechanism.
Recabal A, Caprile T and García-Robles MA (2017). Front. Neurosci. 11:190.
doi: 10.3389/fnins.2017.00190
Neurogenesis in the Hypothalamus of Adult Mice: Potential Role in Energy Balance
Maia V. Kokoeva, Huali Yin, Jeffrey S. Flier. SCIENCE VOL 310 28 OCTOBER 2005
2.- Another topic they could briefly describe is how tickling induces neurogenesis since tickling is a social stimulus.
New insights into the relationship of neurogenesis and affect: tickling induces hippocampal cell proliferation in rats emitting appetitive 50-kHz ultrasonic vocalizations. Behavioural Neuroscience | Research PaperVolume 163, Issue 4p1024-1030November 10, 2009
Author Response
The review “Linking Adult Neurogenesis to Social Reproductive Stimuli: Mechanisms and Functions” is interesting and covers different topics, especially those in which the authors have several publications. I only have a couple of suggestions that the authors need to include, especially the one describing a neurogenic site in the hypothalamus.
We are grateful to the reviewer for his helpful suggestions, which have been incorporated into the text. Below is a point-by-point response to the comments.
Comment 1.- In the introduction, the authors described the 2 classic neurogenic niches: The svz olfactory bulbs and the SVZ/SGZ of the dentate gyrus of the hippocampus. However, evidence also suggests that the hypothalamus is another site of neurogénesis. This third site needs to be included in the review. Here are some references about the topic. Hypothalamic Subependymal Niche: A Novel Site of the Adult Neurogenesis. Ewa Rojczyk-GoÅ‚eË›biewska, Artur PaÅ‚asz, Ryszard Wiaderkiewicz Cell Mol Neurobiol (2014) 34:631–642 DOI 10.1007/s10571-014-0058-5; Hypothalamic Neurogenesis as an Adaptive Metabolic Mechanism. Recabal A, Caprile T and García-Robles MA (2017). Front. Neurosci. 11:190. doi: 10.3389/fnins.2017.00190; Neurogenesis in the Hypothalamus of Adult Mice: Potential Role in Energy Balance Maia V. Kokoeva, Huali Yin, Jeffrey S. Flier. SCIENCE VOL 310 28 OCTOBER 2005.
Response 1. We agree with the reviewer that adult neurogenesis in the hypothalamus should be cited in the review. Following the reviewer's suggestions, we have added new text in the revised version of the Introduction and Cap. 5.1. All changes are highlighted in red in the text.
Introduction
lines 34-37..."Additionally, compelling evidence indicates that the adult mammalian hypothalamus is also a neurogenic area, capable of producing new neurons in physiological conditions, albeit at a much lower rate than the SVZ-OB and the DG (Rojczyk-Gołębiewska et al. 2014) ."
lines 46-48..."For example, hypothalamic neurogenesis is thought to function as an adaptive mecha-nism in response to metabolic changes driven by environmental or internal conditions (Recabal et al. 2017; Kokoeva et al. 2005) "..
Cap.5.1
lines 283-287..." Some of these hormones not only influence adult neurogenesis in the SVZ-OB but also act as potent modulators of neurogenesis in the hippocampal niche (Wan et al. 2021) and in the hypothalamus (Yoo and Blackshaw, 2018). Enhanced neurogenesis at these sites may contribute to learning, memory, mood and other neurophysiological aspects involved in reproductive activities."
2.- Another topic they could briefly describe is how tickling induces neurogenesis since tickling is a social stimulus. New insights into the relationship of neurogenesis and affect: tickling induces hippocampal cell proliferation in rats emitting appetitive 50-kHz ultrasonic vocalizations. Behavioural Neuroscience | Research PaperVolume 163, Issue 4p1024-1030November 10, 2009
Response 2. These are indeed interesting data, which have been included in the revised version of the manuscript in Cap. 5.1
Lines 276-280..."As social interactions are multisensory, modulation of hippocampal neurogenesis may be mediated by other sensory channels in addition to olfaction. For example, in rats, a specific tactile stimulation, tickling, promotes proliferation in the hippocampus when the animal emits a high number of 50 kHz calls, i.e. a condition related to an appetitive state mediated by affective behavior (Wöhr et al. 2009).
Reviewer 3 Report
Comments and Suggestions for Authors
The manuscript “Linking adult neurogenesis to social reproductive mechanisms and functions” by Bovetti et al provides some of the most important insights into what is known in neurogenesis regarding reproductive behavior. The manuscript is well written and easy to follow. The topic is indeed very interesting.
In the abstract, authors propose to lead us to the understanding of the role of adult neurogenesis and its participation in reproductive behavior. In the introduction in turn, authors provide a little history of the first discoveries on neurogenesis and describe the general process of neurogenesis that occurs in two of the most recognized and studied neurogenic regions (since there is also the hypothalamic region that is closely linked to hormonal changes): the olfactory system and the hippocampus. After the introduction, the manuscript remains focused on olfactory neurogenesis, with few examples of the advances of the hippocampal neurogenesis participation in reproduction. It would be appreciated if authors could provide more information regarding the role of hippocampal neurogenesis in the social-reproductive context. It would be very interesting also if some comparisons on the olfactory and hippocampal neurogenesis or discussion could be added to this regard. By adding this information, the manuscript will be very nicely complemented by giving the reader a more general overview of the adult neurogenesis linked to social reproductive stimuli.
Figure. Since hippocampal neurogenesis is also mentioned and described, I believe adding information of this in the Figure would be appropriate. Additionally, in the Figure, the HPG is not found in the abbreviations part. Some spaces are needed in the figure legend, lines 89, 90, 92, 93.
Line 189: extra space
Line 245: extra space
Line 325: extra space
Author Response
The manuscript “Linking adult neurogenesis to social reproductive mechanisms and functions” by Bovetti et al provides some of the most important insights into what is known in neurogenesis regarding reproductive behavior. The manuscript is well written and easy to follow. The topic is indeed very interesting.
We would like to thank the reviewer for his positive comments and helpful suggestions on the manuscript. Below is a point-by-point response to the reviewer's comments.
Comment 1.
In the abstract, authors propose to lead us to the understanding of the role of adult neurogenesis and its participation in reproductive behavior. In the introduction in turn, authors provide a little history of the first discoveries on neurogenesis and describe the general process of neurogenesis that occurs in two of the most recognized and studied neurogenic regions (since there is also the hypothalamic region that is closely linked to hormonal changes): the olfactory system and the hippocampus. After the introduction, the manuscript remains focused on olfactory neurogenesis, with few examples of the advances of the hippocampal neurogenesis participation in reproduction. It would be appreciated if authors could provide more information regarding the role of hippocampal neurogenesis in the social-reproductive context. It would be very interesting also if some comparisons on the olfactory and hippocampal neurogenesis or discussion could be added to this regard. By adding this information, the manuscript will be very nicely complemented by giving the reader a more general overview of the adult neurogenesis linked to social reproductive stimuli.
Response 1. Following the reviewer's comments, we realized that in the submitted version of the manuscript we had not made it clear since the beginning that the focus of the review was to be on olfactory neurogenesis. We have therefore slightly modified the title and text to make this clearer (changes highlighted in red in the revision version of the manuscript). In parallel, we added some more information on DG neurogenesis compared to OB neurogenesis (lines: 273-287), and we modified the figure to introduce hippocampal neurogenesis -and hypothalamic neurogenesis as well- in the drawing. However, we preferred to maintain the original structure of the review maintaining the main focus on olfactory neurogenesis. We believe that the added text and figure modifications enhance the manuscript’s broader view while retaining its focus.
Comment 2. Figure. Since hippocampal neurogenesis is also mentioned and described, I believe adding information of this in the Figure would be appropriate. Additionally, in the Figure, the HPG is not found in the abbreviations part. Some spaces are needed in the figure legend, lines 89, 90, 92, 93.
Line 189: extra space; Line 245: extra space; Line 325: extra space
Response 2. As mentioned above, we have modified the figure to include hippocampal neurogenesis in the schematic in Figure 1B, as suggested by the reviewer. We have also added the hypothalamic neurogenic niche in the same schematic. The figure legend has been changed accordingly and all abbreviations should now be listed in the legend in their full form. Minor corrections have been made according to reviewer's suggestion.